# Exosomal miRNA Profile in Small-for-Gestational-Age Children: A Potential Biomarker for Catch-Up Growth

**DOI:** 10.3390/genes13060938

**Published:** 2022-05-24

**Authors:** Hwal Rim Jeong, Jae-A Han, Heeji Kim, Hye Jin Lee, Young Suk Shim, Min Jae Kang, Jong Seo Yoon, Seongho Ryu, Il Tae Hwang

**Affiliations:** 1Department of Pediatrics, College of Medicine, Soonchunhyang University, Cheonan 31151, Korea; hrjeong@schmc.ac.kr; 2Soonchunhyang Institute of Medio-Bio Science (SIMS), Soonchunhyang University, Cheonan 31151, Korea; flybird34@gmail.com (J.-A.H.); khjee1932@naver.com (H.K.); 3Department of Pediatrics, College of Medicine, Hallym University, Chuncheon 24252, Korea; hjleeped@hallym.or.kr (H.J.L.); remoni80@gmail.com (M.J.K.); yjs1026@kdh.or.kr (J.S.Y.); 4Department of Pediatrics, School of Medicine, Ajou University, Suwon 16499, Korea; royjays@aumc.ac.kr

**Keywords:** SGA, catch-up growth, miRNA, exosome

## Abstract

**Objective**: The mechanism underlying postnatal growth failure and catch-up growth in small-for-gestational-age (SGA) children is poorly understood. This study investigated the exosomal miRNA signature associated with catch-up growth in SGA children. **Methods**: In total, 16 SGA and 10 appropriate-for-gestational-age (AGA) children were included. Serum exosomal miRNA was analyzed using next-generation sequencing (NGS). Exosomal miRNA was profiled for five SGA children with catch-up growth (SGA-CU), six SGA children without CU growth (SGA-nCU), and five AGA children. **Results**: Exosomal miRNA profiles were clustered into three clear groups. The exosomal miRNA expression profiles of the SGA-nCU group differed from those of the SGA-CU and AGA groups. In all, 22 miRNAs were differentially expressed between SGA-nCU and AGA, 19 between SGA-nCU and SGA-CU, and only 6 between SGA-CU and AGA. In both SGA-nCU and SGA-CU, miR-874-3p was upregulated and miR-6126 was downregulated. Therefore, these two miRNAs could serve as biomarkers for SGA. Compared with SGA-CU and AGA, miR-30c-5p, miR-363-3p, miR-29a-3p, and miR-29c-3p were upregulated in SGA-nCU, while miR-629-5p and miR-23a-5p were downregulated. These six miRNAs could be associated with growth failure in SGA-nCU children. **Conclusions**: SGA children without CU have a distinct exosomal miRNA expression profile compared with AGA and SGA children with CU. Exosomal miRNAs could serve as novel biomarkers for CU.

## 1. Introduction

Small for gestational age (SGA) is defined as birthweight and/or height below 2.0 standard deviations (SDs) for gestational age, which is approximately below the third percentile [1]. It is also defined as birthweight below the 10th percentile for gestational age [2]. The causes of SGA are multifactorial, including maternal, placental, and demographic factors. Genetic abnormalities can also cause SGA [3,4,5]. Imprinting disorders; altered methylation, such as in Silver–Russel syndrome; chromosome abnormalities such as in Turner syndrome; and copy number variation (CNV) are associated with genetic causes of SGA. Growth hormone and insulin-like growth factor (GH-IGF) axis disorders such as *IGF1R* mutations, genetic defects in growth plate regulation, and paracrine factors are also associated with SGA. In 2012, the prevalence of SGA was 19.3% in low- and middle-income countries, with the greatest prevalence in South Asia [6]. In 2010–2011, it was 13.4% in South Korea [7]. SGA children are at increased risk for growth attenuation, precocious puberty, type-2 diabetes, metabolic syndrome, obesity, and psychological impairment [8,9,10,11].

Catch-up growth (CU) is defined as growth (cm/y) greater than the median for chronologic age and sex [12]. The majority of SGA children show spontaneous CU during the first years of life and attain a height above –2 SD by the age of 2 years. However, approximately 10% fail to show sufficient CU and continue to have short stature throughout childhood and adolescence [13,14]. Therefore, SGA accounts for approximately 20% of all cases of short stature [15]. Numerous genetic, environmental, hormonal, and nutritional factors may affect CU in SGA [16,17,18]. However, the mechanism of postnatal CU in SGA children is unclear. It is difficult to predict which SGA children will achieve CU.

MicroRNAs (miRNAs) are small, noncoding single-stranded RNAs that are 18–25 nucleotides in length and are highly conserved across species. They are involved in posttranscriptional regulation of gene expression, and have important regulatory roles in most biological processes [19]. Circulating miRNA profiles differ between physiological and pathological states. Dysregulated miRNA expression is associated with several pathologies, including cancers [20], metabolic syndrome [19], and neurodegenerative and chronic inflammatory diseases [21]. Thus, miRNAs have emerged as potential therapeutic targets and diagnostic markers for various diseases [22].

Various miRNAs circulate in body fluids but they are also circulated by heterogeneous carriers, such as exosomes, microparticles, and apoptotic vesicles. Exosomes transport proteins, lipids, DNA, RNA, and miRNA between cells. The miRNAs contained in exosomes are particularly stable in biological fluids because they are protected by a double lipid layer. Moreover, exosomes contain specific repertoires of miRNAs, which are selectively sorted in these vesicles. Therefore, the expression patterns of exosomal miRNAs differ between healthy and diseased individuals; thus, their profiles may provide important clues to unravel the pathophysiology of various diseases.

Several recent studies have implied a strong association between circulatory miRNAs and impaired growth and metabolism in SGA children [23,24]. The role of exosomal miRNAs in CU in SGA children remains unknown. In this study, we investigated differentially expressed exosomal miRNAs and used miRNA sequencing and bioinformatic analysis to explore their potential roles in postnatal growth in SGA children.

## 2. Materials and Methods

### 2.1. Study Subjects

Sixteen SGA children and ten age-matched appropriate for gestational age (AGA) children were enrolled. Clinical data were collected retrospectively via chart review of patients in the pediatric endocrinology clinic at the Hallym Medical Center, Seoul, Republic of Korea, between 1 January 2013 and 31 December 2018. The subjects were divided into three groups. SGA was defined as birthweight at least below the third percentile for the corresponding gestational age and sex. Most SGA children achieve spontaneous CU and reach a height above –2 SD by the age of 2 years. Therefore, we hypothesized that SGA children would have completed sufficient catch-up growth after the age of four. In this study, SGA with CU (SGA-CU) was defined as SGA children whose height was more than that of the 10th percentile for their age and sex after the age of four years. SGA without catch-up (SGA-nCU) was defined as SGA children whose height remained less than the third percentile after the age of 4 years. The healthy control group included AGA children whose height was in the 10th–90th percentiles after the age of 4 years. All subjects were born at gestational ages of 37–42 weeks from singleton pregnancies. Patients with chronic diseases, developmental disorders, prematurity (gestational age < 37 weeks), large for gestational age (birthweight ˃ 90th percentile), and congenital malformations were excluded.

Data on birth history, height, weight and body mass index (BMI) were collected from clinical charts and electronic medical records. BMI was calculated as body mass (kg) divided by height squared (m^2^). Birthweight percentile was analyzed using the Korean reference for birthweight for gestational age and sex [25]. SDs were calculated using the least mean square (LMS) method of the 2017 Korean National Growth Charts. The study protocol was approved by the Institutional Review Board of Soonchunhyang University Hospital (IRB No. 2019-11-039), and informed consent was obtained from all patients or their parents before enrollment.

### 2.2. Clinical Characteristics of the Study Subjects

In total, 16 SGA children and 10 healthy controls were included in this study. Among the SGA children, 5 were SGA-CU and 11 were SGA-nCU. Ten subjects (5 AGA and 5 SGA-nCU) were analyzed based on circulating serum miRNA NGS, with the AGA group serving as a control. Table 1 shows the clinical characteristics of the subjects whose miRNAs were analyzed. According to birth records, the gestational ages of the SGA-nCU and control groups were similar, but birthweights were significantly lower in the SGA-nCU group. The height and weight standard deviation scores (SDSs) were significantly lower in the SGA-nCU group than in controls.

The remaining five, six, and five subjects from the SGA-CU, SGA-nCU, and AGA groups, respectively, were analyzed using serum exosomal miRNA. Their clinical data are shown in Table 2. According to birth records, these groups had similar gestational ages, but their birthweights differed significantly.

### 2.3. Sample Collection

Blood samples were centrifuged, and serum was immediately stored at –80 °C until analyses. Then miRNAs were extracted from both serum and serum exosomes. For serum, miRNA differential expression was analyzed only between the SGA-nCU and AGA groups. For the miRNA derived from serum exosomes, SGA-CU, SGA-nCU, and AGA were all compared.

### 2.4. Exosome Isolation from Human Serum

ExoQuick exosome precipitation solution (System Biosciences, Palo Alto, CA, USA) was used to isolate exosomes from 1-mL serum samples according to the manufacturer’s instructions. Briefly, serum was centrifuged at 3000× *g* for 15 min to remove cells and debris. Then ExoQuick solution was added and the mixture was incubated at 4 °C. Samples were centrifuged at 1500× *g* for 30 min at room temperature, and the resulting pellets were resuspended in 100 µL of phosphate-buffered saline (PBS). miRNAs were extracted from exosomes using an miRNeasy Mini Kit (QIAGEN, Hilden, Germany) according to the recommended protocol, and then eluted in 20 µL RNase-free water.

### 2.5. Serum and Exosomal miRNA Next-Generation Sequencing (NGS)

Patient serum and exosomal miRNAs were processed, and 10 ng was used as an input for each library. Small RNA libraries were constructed using the SMARTer smRNA-Seq Kit (Takara Bio, Shiga, Japan) according to the manufacturer’s instructions. Sequencing libraries were generated by polyadenylation, complementary DNA (cDNA) synthesis, and polymerase chain reaction (PCR) amplification.

The libraries were gel-purified and validated by assessing size, purity, and concentration using an Agilent 2100 Bioanalyzer (Agilent Technologies, Santa Clara, CA, USA). Equimolar amounts of the libraries were pooled and sequenced on an Illumina HiSeq 2500 instrument (Illumina, San Diego, CA, USA) to generate 101 base reads. Image decomposition and quality value calculations were performed using the modules of the Illumina pipeline. Clustered reads were aligned to the reference genome from miRBase 21 to identify miRNA.

### 2.6. Differential miRNA Expression Analysis

The reads were normalized according to relative logarithmic expression using DESeq2 (Genome Biology Unit; European Molecular Biology Laboratory, Heidelberg, Germany). For each miRNA, base mean and log-fold change values were calculated and compared among the groups. A statistical hypothesis test was conducted to compare the groups using the negative binomial Wald test in DESeq2. Differentially expressed miRNAs between the two groups were determined by assessing miRNAs with a |fold change|  ≥  2 and false discovery rate-adjusted *p*-values  <  0.05. We also performed hierarchical clustering analysis using complete linkage and Euclidean distance as measures of similarity to display the expression patterns of differentially expressed miRNAs that satisfied the criteria mentioned above. To visualize the dot plots for RPM values, the R package grammar of graphics (gg) was used.

### 2.7. Statistical Analysis

Statistical analysis was performed using the SPSS Statistics software (v. 20.0; IBM Corp., Armonk, NY, USA). Comparisons between two groups were performed using the Mann–Whitney U test. Comparisons among three groups were performed using the Kruskal–Wallis test. A *p*-value < 0.05 was considered statistically significant.

## 3. Results

### 3.1. Profile of Circulating Serum miRNA Expression in SGA-nCU and AGA

Serum miRNAs from SGA-nCU and AGA were subjected to miRNA sequencing. These two groups did not cluster together, as depicted by the heat map (Figure 1). In other words, serum miRNA could not clearly distinguish between the two groups. In total, 72 serum miRNAs were differentially expressed, of which 43 were upregulated and 29 were downregulated. Table 3 shows the fold-change values of the top 20 differentially expressed miRNAs between the two groups.

### 3.2. Exosomal miRNA Profiles Show Different Expression Patterns than Those Collected from Serum

In exosomal miRNA sequencing, the SGA-nCU and AGA groups clustered together, with clear divisions (Figure 2). We identified a total of 22 miRNAs that were differentially expressed in SGA-nCU, meeting the criteria of fold > |2| and *p* < 0.05. However, miRNAs with read counts of less than 10 in all samples were removed. In total, 7 upregulated and 10 downregulated miRNAs differed significantly between SGA-nCU and AGA (Table 4). There were significantly fewer differentially expressed exosome-derived miRNAs, and only three of them were also found in serum (has-miR-29a-3p, has-miR-29c-3p and has-miR-15a-5p).

### 3.3. Exosomal miRNA Profiles of SGA-CU Were More Similar to Those of AGA than Those of SGA-nCU

Despite the fact that SGA-CU and AGA formed two clear groups in clustering analysis, as shown in the heat map (Figure 3), only six miRNAs were differentially expressed (Table 5a). Compared with AGA, miR-874-3p was upregulated and miR-6126 was downregulated in both SGA-nCU and SGA-CU (Table 5b). Thus, these two miRNAs are potential markers for the SGA trait.

### 3.4. Exosomal miRNA Profiles of SGA-nCU and SGA-CU Show Potential Differences

The groupings for SGA-nCU and SGA-CU were defined by a marginal difference in one SGA-CU sample (Figure 4). In total, 19 miRNAs were differentially expressed (Table 6a). Notably, six miRNAs shared the same trend when SGA-nCU was compared with AGA. The miRNAs miR-30c-5p, miR-363-3p, miR-29a-3p, and miR-29c-3p were upregulated, and miR-629-5p and miR-23a-5p were downregulated (Table 6b). These miRNAs may potentially be biomarkers for predicting whether SGA children will show CU; however, further studies are needed to validate this and elucidate the mechanisms.

### 3.5. Comparison of the Exosomal miRNA Profiles of AGA, SGA-CU, and SGA-nCU Children

To compare the AGA, SGA-CU, and SGA-nCU groups, we averaged the RPM of each miRNA by group, and plotted a graph according to their percent expression (Figure 5). Then these RPM values were compared with the actual differential expression, and miRNAs with fold > |2| and *p* < 0.05 were analyzed. In the AGA group, miR-6126 had significantly greater expression, while miR-874-3p had significantly lower expression compared with SGA-CU and SGA-nCU. Changes in the expression of these miRNAs were common in both SGA-CU and SGA-nCU, and could help determine the SGA trait. In SGA-nCU, four signature miRNAs (miR-29a-3p, miR-29c-3p, miR-30c-5p, and miR-363-3p) were overexpressed and two signature miRNAs (miR-23a-5p and miR-629-5p) were under-expressed compared with SGA-CU and AGA. These miRNAs could be related to the mechanism of CU in SGA children.

## 4. Discussion

This study presents the exosomal miRNA profile that is differentially expressed in SGA children with CU, without CU, and AGA children with normal growth. The two miRNAs miR-874-3p and miR-6126 may be biomarkers for SGA. In addition, six miRNAs may predict CU in SGA children. This study provides evidence that exosomal miRNAs may be associated with the SGA trait and CU in SGA children. To the best of our knowledge, this is the first study of the exosomal miRNA profile of SGA children.

SGA children without CU have short statures even after reaching adulthood. Moreover, postnatal CU in SGA children is important with respect to the risk for future metabolic disease. To date, little has been known about the underlying mechanism of CU and this increased metabolic risk, making it difficult to predict which SGA children will grow properly or develop metabolic diseases.

Mas-Parés et al. [23] studied umbilical-cord miRNA profiles in SGA-CU, SGA-nCU, and AGA children, and their association with CU. They reported 12 differentially expressed miRNAs between SGA-CU and SGA-nCU at 12 months and 6 years of age. After validation, they found that umbilical cord miR-576-5p was associated with postnatal growth and cardiometabolic risk in SGA children, and they suggested its use as a novel biomarker for early identification of CU in SGA infants. In our study, miR-576-5p expression differed significantly between the SGA-CU and SGA-nCU groups. Meanwhile, differences in miR-576-5p expression between SGA-nCU and AGA and between SGA-CU and AGA were not significant (Appendix A). Our study had a cross-sectional design, and all of the subjects were healthy, except for the short statures in the SGA-nCU group; none had any signs of any cardiometabolic diseases. Therefore, the role of the miR-575-5p in CU and cardiometabolic complications in SGA requires validation in further studies.

Inzaghi et al. [26] evaluated serum miRNA levels longitudinally in 23 SGA and 27 AGA subjects at the ages of 9 and 21 years. There were no differences in circulating miRNA levels between the two groups. In SGA children, miR-122-5p expression at 9 years was inversely related to adiponectin levels at 21 years, implying that miR-122-5p could be used to identify SGA children at a higher risk for metabolic dysfunction. In our study, miR-122-5p did not differ significantly among the three groups. In addition, our study included younger subjects, and cardiometabolic assessment, including the evaluation of lipid profiles, was not performed.

Our study was performed in different age groups and used exosomal miRNA; therefore, our findings differ significantly from those of previous studies. We initially compared SGA-nCU and AGA groups using miRNA extracted from whole serum. However, despite a large list of significantly expressed miRNAs, they did not cluster clearly into two groups. This implies the limitations of using whole-serum miRNA for high-throughput analysis.

Next, we took an in-depth approach and used exosomal miRNAs to further investigate the expression patterns in the two groups. Surprisingly, SGA-nCU and AGA clustered into two definite groups. The drawback of serum exosomal miRNA-sequencing, however, is that the relative scarcity of exosomal miRNA leads to fewer miRNA reads than are found in whole serum, and hence there are fewer significantly and differentially expressed miRNAs. In this study, 72 miRNAs were differentially expressed in whole-serum miRNA sequencing, while 17 were differentially expressed in exosomal miRNA sequencing. The miRNAs specifically isolated from exosomes were important for understanding and characterizing the differences between these patient groups. Serum exosomal miRNA sequencing distinguished SGA-nCU from both SGA-CU and AGA. Exosomes effectively removed the heterogeneity within the sample groups, but only to the extent that they were distinguishable between any two given sample groups.

Notably, SGA-nCU was as different from SGA-CU as it was from AGA, showing a similar pattern of differentially expressed miRNAs. We found that miR-30c-5p, miR-363-3p, miR-29a-3p, and miR-29c-3p were highly expressed compared with both AGA and SGA-CU, while miR-629-5p and miR-23a-5p were downregulated. These miRNAs have not often been found in studies using circulatory miRNAs [24,26].

In renal cell carcinoma, miR-30c-5p expression is significantly reduced. This adversely affects cell proliferation whereas overexpression suppresses tumor formation [27,28,29]. The miRNA miR-363-3p is associated with various diseases and its overexpression decreases cell proliferation [30,31,32]. Recently, miR-363-3p was shown to have suppressive effects on neonatal hypoxic ischemic encephalopathy [33]. In addition, miR-363-3p suppresses tumor growth and miR-363-3p inhibition facilitates cell proliferation [34]. Downregulated miR-29a-3p expression inhibits cell proliferation in many different types of cancers. In particular, it binds to, and directly regulates, the insulin-like growth factor 1 receptor (IGF1R) gene in hepatocellular carcinoma [35]. The IGF1R mutation is an etiology of SGA. Increased miR-29a-3p expression leads to downregulation of IGF1R expression, which may be related to the poor growth in the SGA-nCU group. The miRNA miR-29c-3p has been associated with various cancers [36,37,38], and is upregulated in congenital heart disease. It is also involved in promoting the senescence of human mesenchymal stem cells (hMSCs), and downregulation of miR-29c-3p inhibits hMSC differentiation [39]. Overexpression of miR-629-5p is involved in the processes of tumor cell growth, invasion, and metastasis in hepatocellular carcinoma [40], lung cancer [41], colorectal cancer [42], ovarian cancer [43] and pancreatic cancer [44]. The miRNA miR-23a-5p is involved in regulating normal growth, differentiation, and apoptosis of cells, together with various physiological and pathological processes [45,46,47]. It is also involved in multiple cancers, including pancreatic cancer cells [48], acute myeloid leukemia cells [49], renal cell carcinoma cells [50], and rheumatoid arthritis synovial fibroblasts [51]. In addition, exosomal miR-23a-5p may also be involved in bone remodeling [52]. Altogether, upregulated miRNAs were found to inhibit cell proliferation, while downregulated miRNAs were found to have an oncogenic role. SGA children may be born small because upregulated tumor suppressor miRNA potentially suppresses early child growth, while downregulated oncogenic miRNA may also play a critical role.

Based on exosomal miRNA expression profiles, SGA-CU was more similar to AGA than was SGA-nCU. This information is significant, as it implies that although SGA-CU children are characterized as small at birth, they are actually closer to healthy infants compared with SGA-nCU children. In both SGA-CU and SGA-nCU, miR-874-3p was upregulated and miR-6126 was significantly downregulated. Thus, these miRNAs could serve as potential biomarkers for the SGA trait in children. However, studies of these miRNAs have not reported consistent trends. For example, miR-874-3p has been reported to inhibit cell proliferation, while miR-6126 has been found to be a tumor suppressor in ovarian cancer [53,54]. This may be due to the fewer numbers of studies with these miRNAs, but our results imply that the SGA trait is difficult to characterize through a comparison of SGA-nCU and SGA-CU with AGA. In other words, the miRNA profile of SGA-CU is actually closer to AGA after CU until the age of 4 years.

An overall comparison of the three groups by RPM of the expressed miRNAs (Figure 5) showed that the SGA trait was signified by the loss of miR-6126 expression and increased miR-874-3p expression. The SGA-CU group, which achieved CU, showed miRNA expression patterns similar to those of the AGA group, and no significant differences in miRNA expression were observed in the SGA-CU group compared with the AGA and SGA-nCU groups. SGA-nCU showed a total of six differentially expressed miRNAs, four and two of which showed increased and decreased expression patterns, respectively. Conversely, it could be inferred that four downregulated miRNAs and two upregulated miRNAs are involved in catch-up growth in SGA children (Figure 6). These miRNAs are involved in cell proliferation, cell growth, and IGF1 receptor regulation and may potentially serve as markers for the growth potential in SGA children.

This study was a cross-sectional study with relatively few subjects, which makes it difficult to generalize the results to all SGA patients. Further validation studies with more subjects are needed. Longitudinal, prospective studies from birth to adulthood are needed to apply our findings to predicting catch-up growth in babies born SGA. Although the mechanism underlying catch-up growth remains unclear, we identified miRNAs involved in catch-up growth, which may help clarify the processes involved. Future studies to elucidate the downstream pathways of these miRNAs may lead to the discovery of novel target genes that are essential for child growth. Despite several limitations, to the best of our knowledge, this is the first study to illustrate the expression patterns and clinical significance of exosomal miRNAs in SGA children. SGA children without CU have a distinct exosomal miRNA expression profile compared with AGA and SGA children with CU. Our results imply that exosomal miRNA profiling is a better way of classifying SGA than serum miRNA profiling. Exosomal miRNAs may serve as potential prognostic biomarkers and aid in the development of novel therapeutic strategies for CU in SGA children.

## Figures and Tables

**Figure 1 genes-13-00938-f001:**
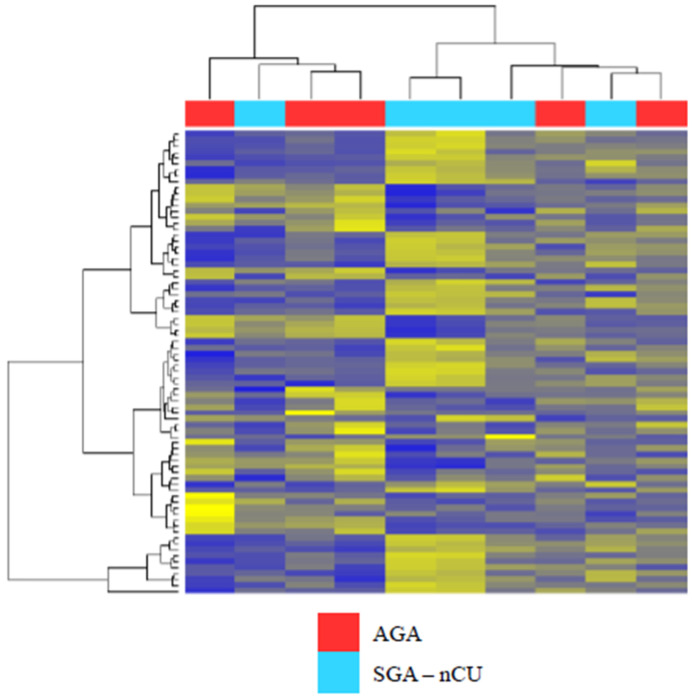
Heat map of the Z-scores of serum miRNAs for the SGA-nCU (n = 5) and AGA (n = 5) groups revealing that serum miRNA expression did not cluster in the two groups. Blue: Z score < −2.0. Yellow: Z score > 2.0.

**Figure 2 genes-13-00938-f002:**
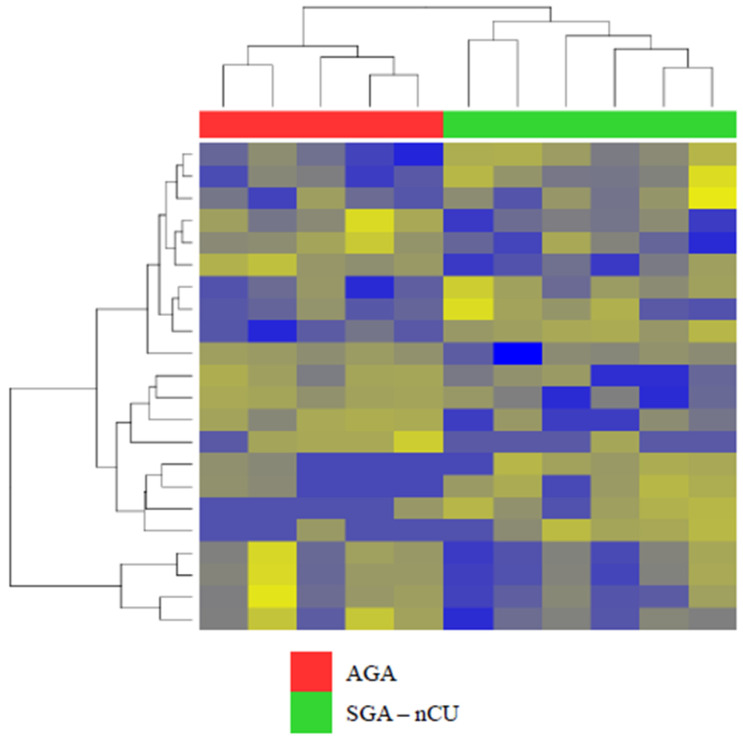
Heat map of the Z-scores of exosomal miRNAs for the SGA-nCU (n = 6) and AGA (n = 5) groups.

**Figure 3 genes-13-00938-f003:**
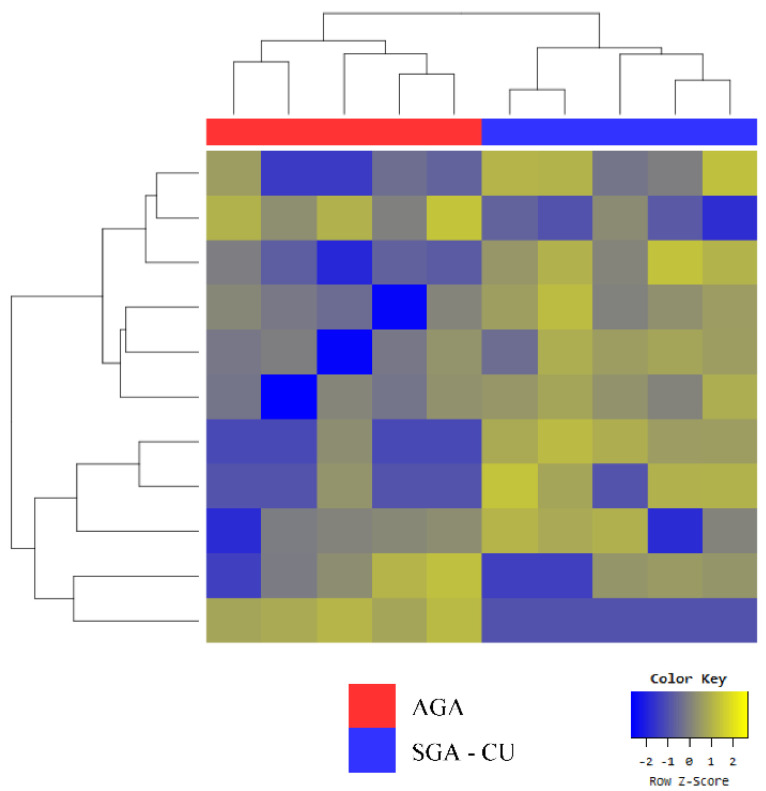
Heat map of the Z-scores of exosomal miRNAs for the SGA-CU (n = 5) and AGA (n = 5) groups.

**Figure 4 genes-13-00938-f004:**
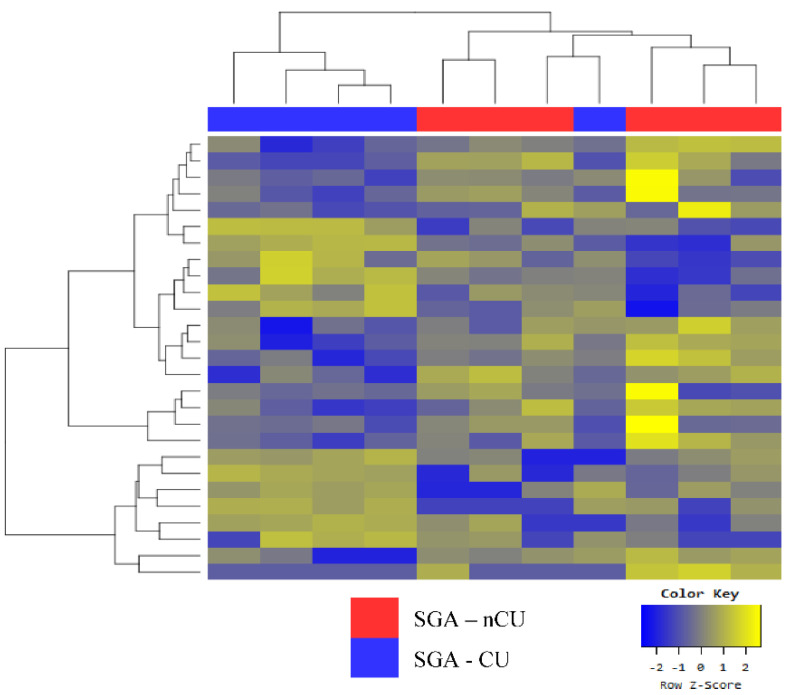
Heat map of the Z-scores of exosomal miRNAs for the SGA-nCU (n = 6) and SGA-CU (n = 5) groups.

**Figure 5 genes-13-00938-f005:**
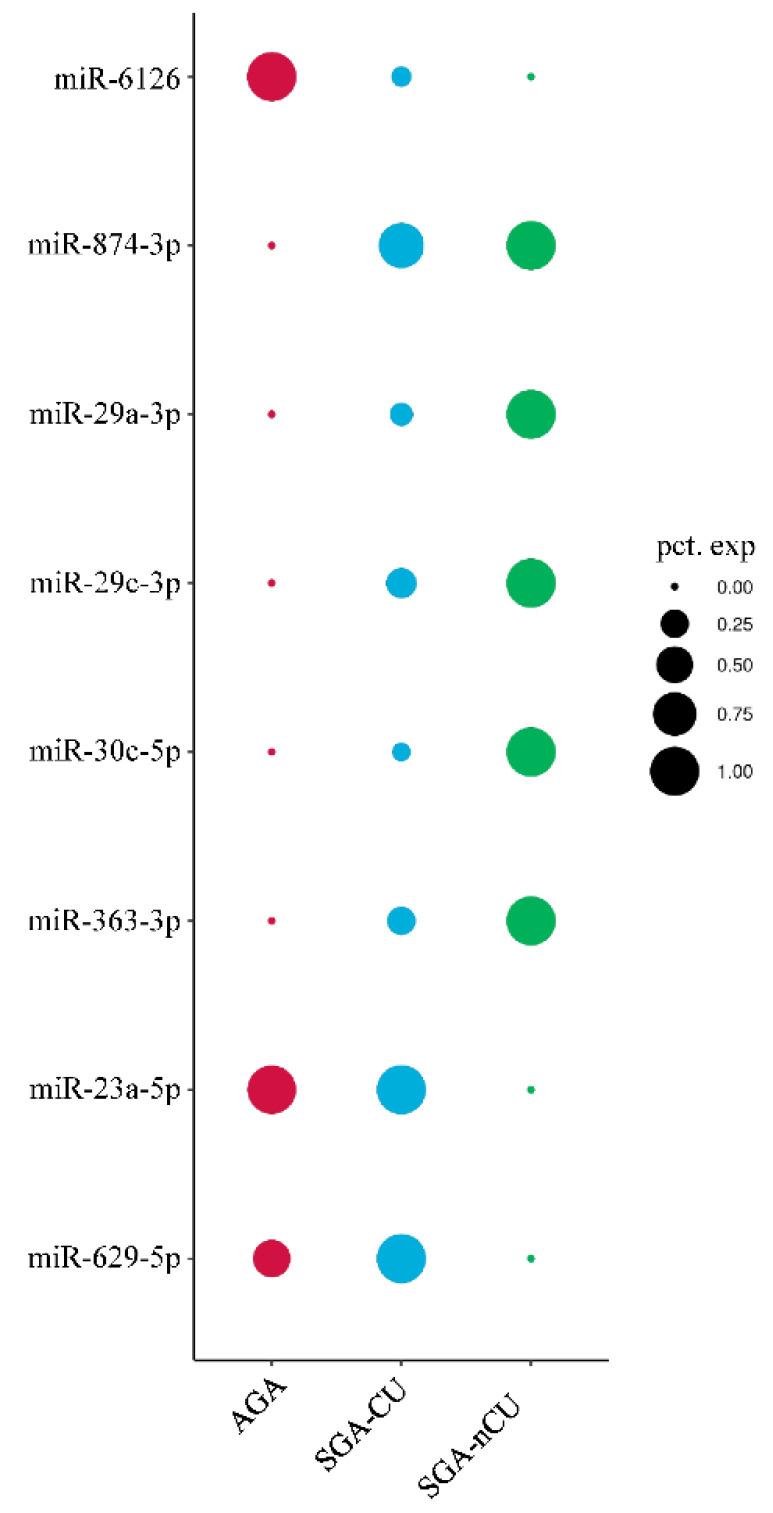
Schematic diagram of the average RPM percent expression of each miRNA in all groups.

**Figure 6 genes-13-00938-f006:**
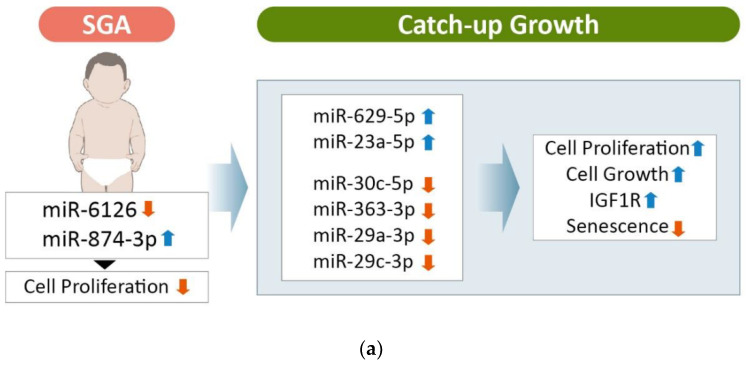
Candidate miRNAs involved in catch-up growth in SGA children. (**a**) These miRNAs are presumed to be involved in catch-up growth though regulation of cell proliferation, cell growth, IGF1 receptor and aging process. (**b**) Change in height and associated miRNAs with catch-up growth in SGA-CU and SGA-nCU children.

**Table 1 genes-13-00938-t001:** Clinical characteristics of the subjects who underwent serum miRNA analysis.

Total (n = 10)	SGA-nCU	AGA	*p*-Value
n (%)	5 (50%)	5 (50%)	-
Boys/girls	3/2	2/3	-
GA (weeks)	38.80 ± 1.64	39.40 ± 1.34	0.548
Birth weight (kg)	2.44 ± 0.18	3.34 ± 0.27	0.008
CA (year)	4.54 ± 0.28	4.63 ± 0.28	0.690
Height (cm)	97.22 ± 1.96	106.94 ± 1.66	0.008
Height SDS	–2.17 ± 0.19	0.15 ± 0.45	0.008
Weight (kg)	13.62 ± 1.62	17.30 ± 1.83	0.008
Weight SDS	–2.51 ± 1.01	–0.28 ± 0.79	0.008
BMI SDS	–1.26 ± 1.15	–0.62 ± 1.15	0.690

CU, catch-up growth; GA, gestational age; CA, chronological age; SDS, standard deviation score; BMI, body mass index. The Mann–Whitney U test was performed.

**Table 2 genes-13-00938-t002:** Clinical characteristics of SGA and control cases who underwent miRNA analysis using exosomes.

Total (n = 16)	SGA		AGA	*p*-Value
	SGA-nCU	SGA-CU		
n (%)	6 (37.5)	5 (31.25)	5 (31.25)	-
Boys/girls	3/3	2/3	2/3	-
GA (weeks)	38.60 ± 1.36	38.20 ± 1.09	39.20 ± 1.30	0.353
Birth weight (kg)	2.39 ± 0.24	2.06 ± 0.23	3.47 ± 0.29	0.003
CA (year)	4.44 ± 0.31	5.93 ± 0.77	5.87 ± 1.06	0.018
Height (cm)	96.66 ± 1.89	111.66 ± 3.78	117.58 ± 7.92	0.004
Height SDS	–2.10 ± 0.27	–0.61 ± 0.39	0.69 ± 0.82	0.001
Weight (kg)	14.16 ± 1.74	17.96 ± 1.88	25.50 ± 7.42	0.003
Weight SDS	–2.06 ± 1.14	–1.17 ± 0.67	1.19 ± 1.15	0.006
BMI SDS	–0.62 ± 1.21	–1.17 ± 0.85	1.12 ± 1.52	0.025

The Kruskal–Wallis test was performed. CU, catch-up growth; GA, gestational age; BW, birthweight; CA, chronological age; BMI, body mass index; SDS, standard deviation score.

**Table 3 genes-13-00938-t003:** Fold change in miRNA expression in the SGA-nCU group compared with the AGA group.

miRNA	Fold Change SGA-nCU/AGA	*p*-Value
**hsa-miR-29b-3p**	7.03	<0.001
**hsa-miR-4448**	5.34	0.000
**hsa-miR-141-3p**	4.97	0.001
**hsa-miR-29a-3p**	4.18	0.004
**hsa-miR-144-5p**	4.14	0.003
**hsa-miR-26b-5p**	4.00	0.006
**hsa-miR-29c-3p**	3.93	0.008
**hsa-let-7g-5p**	3.81	0.002
**hsa-miR-32-5p**	3.48	0.011
**hsa-miR-96-5p**	3.45	0.012
**hsa-miR-543**	–2.56	0.031
**hsa-miR-493-3p**	–2.64	0.002
**hsa-miR-125b-1-3p**	–2.66	0.017
**hsa-miR-409-3p**	–2.86	0.013
**hsa-miR-3120-5p**	–2.90	0.030
**hsa-miR-485-3p**	–2.92	0.032
**hsa-miR-758-3p**	–2.98	0.002
**hsa-miR-654-5p**	–3.30	0.002
**hsa-miR-370-3p**	–3.35	0.005
**hsa-miR-127-3p**	–3.49	0.005

**Table 4 genes-13-00938-t004:** Fold change in exosomal miRNA expression in the SGA-nCU compared with the AGA group.

miRNA	Fold Change SGA-nCU/AGA	*p*-Value
**hsa-miR-3651**	11.17	0.006
**hsa-miR-874-3p**	3.65	0.002
**hsa-miR-363-3p**	3.37	0.007
**hsa-miR-29a-3p**	2.44	0.049
**hsa-miR-15a-5p**	2.44	0.043
**hsa-miR-29c-3p**	2.43	0.009
**hsa-miR-30c-5p**	2.09	0.043
**hsa-miR-505-5p**	–2.05	0.039
**hsa-let-7d-5p**	–2.07	0.044
**hsa-miR-629-5p**	–2.24	0.040
**hsa-miR-6126**	–2.35	0.049
**hsa-let-7f-5p**	–2.78	0.033
**hsa-let-7a-5p**	–2.87	0.029
**hsa-let-7e-5p**	–2.93	0.007
**hsa-miR-23a-5p**	–3.25	0.004
**hsa-miR-28-5p**	–4.80	0.002
**hsa-miR-1-3p**	–4.80	0.004

**Table 5 genes-13-00938-t005:** Exosomal miRNA profile of the SGA-CU and AGA groups. (**a**) Fold change in exosomal miRNA expression of the SGA-CU group compared with the AGA group. (**b**) miRNAs with similar expression trends between SGA-nCU and AGA, and between SGA-CU and AGA.

(a)				
miRNA	Fold Change SGA-CU/AGA		*p*-Value	
**hsa-miR-1285-5p**	3.90		0.006	
**hsa-miR-874-3p**	3.29		0.004	
**hsa-miR-339-3p**	3.06		0.027	
**hsa-miR-143-3p**	2.08		0.038	
**hsa-miR-6126**	–2.12		0.027	
**hsa-miR-627-5p**	–4.26		0.044	
(**b**)				
	**SGA-nCU vs. AGA**		**SGA-CU vs. AGA**	
**miRNA**	**Fold Change**	** *p* ** **-Value**	**Fold Change**	** *p* ** **-Value**
**hsa-miR-874-3p**	3.65	0.002	3.29	0.004
**hsa-miR-6126**	–2.35	0.049	–2.12	0.027

**Table 6 genes-13-00938-t006:** Exosomal miRNA profiles of the SGA-nCU and SGA-CU groups. (**a**) Fold change in exosomal miRNA expression of the SGA-nCU group compared with the SGA-CU group. (**b**) miRNAs showing similar expression trends between SGA-nCU and SGA-CU, and between SGA-nCU and AGA.

(a)				
miRNA	Fold Change SGA-nCU/SGA-CU		*p*-Value	
**hsa-miR-342-3p**	3.59		0.010	
**hsa-miR-576-5p**	3.55		0.001	
**hsa-miR-150-5p**	3.10		0.006	
**hsa-miR-192-5p**	2.85		0.001	
**hsa-miR-29b-3p**	2.83		0.006	
**hsa-miR-101-3p**	2.65		0.035	
**hsa-miR-30c-5p**	2.58		0.010	
**hsa-miR-363-3p**	2.45		0.046	
**hsa-miR-29a-3p**	2.38		0.037	
**hsa-miR-25-3p**	2.22		0.011	
**hsa-miR-21-5p**	2.17		0.012	
**hsa-miR-29c-3p**	2.03		0.027	
**hsa-miR-335-5p**	–2.04		0.012	
**hsa-miR-574-5p**	–2.05		0.035	
**hsa-miR-193a-5p**	–2.37		0.035	
**hsa-miR-629-5p**	–2.62		0.012	
**hsa-miR-23a-5p**	–2.85		0.005	
**hsa-miR-382-5p**	–2.88		0.039	
**hsa-miR-326**	–2.91		0.007	
**(b)**				
	**SGA-nCU vs. SGA-CU**		**SGA-nCU vs. AGA**	
**miRNA**	**Fold Change**	** *p* ** **-Value**	**Fold Change**	** *p* ** **-Value**
**hsa-miR-30c-5p**	2.58	0.010	2.09	0.043
**hsa-miR-363-3p**	2.45	0.046	3.37	0.007
**hsa-miR-29a-3p**	2.38	0.037	2.44	0.049
**hsa-miR-29c-3p**	2.03	0.027	2.43	0.009
**hsa-miR-629-5p**	–2.62	0.012	–2.24	0.040
**hsa-miR-23a-5p**	–2.85	0.005	–3.25	0.004

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
