# Peer review of "Exosomal miRNA Profile in Small-for-Gestational-Age Children: A Potential Biomarker for Catch-Up Growth"

_genes, 2022, doi:10.3390/genes13060938_

Round 1
Reviewer 1 Report
My suggestions:
- In the introduction, could you mention a few examples of genetic risk factors for SGA?
- Table 1. would be better to add to Materials and methods, and the Study subjects part.
- Also in Table 1, at GA (gestational age) and CA (chronologic age) , I would months. For example: GA, gestational age (months); BW, birth weight and current weight (kg); CA, chronologic age (months), height (cm)
- Figure 1 may be uploaded in better resolution
- In the Discussion, I would add a figure, which could show, how through which pathways the found miRNAs could play a role in SGA.
- In future this study may be repeated in larger sample size to validate.
Author Response
Q 1. In the introduction, could you mention a few examples of genetic risk factors for SGA?
à Thank you for your useful comments. There are many genetic risk factors for SGA, including various syndromes (like Silver Russel syndrome), genetic defects, and chromosomal abnormalities. We have added simple genetic abnormalities to the Introduction.
“The causes of SGA are multifactorial, including maternal, placental, and demographic factors. Genetic abnormalities can also cause SGA. Imprinting disorders; altered methylation, such as in Silver–Russel syndrome; chromosome abnormalities such as in Turner syndrome; and copy number variation (CNV) are associated with genetic causes of SGA. Growth hormone and insulin-like growth factor (GH-IGF) axis disorders such as IGF1R mutations, genetic defects in growth plate regulation, and paracrine factors are also associated with SGA.”
Q 2. Table 1. would be better to add to Materials and methods, and the Study subjects part.
à As suggested, the clinical characteristic of the subject were moved to the Materials and Methods.
Q 3. Also in Table 1, at GA (gestational age) and CA (chronological age), I would months. For example: GA, gestational age (months); BW, birth weight and current weight (kg); CA, chronologic age (months), height (cm)
à We have added units to each indicator.
Q 4. Figure 1 may be uploaded in better resolution
à We have improved Figure 1.
Q 5. In the Discussion, I would add a figure, which could show, how through which pathways the found miRNAs could play a role in SGA.
à We have added a new Figure 6 and the following to the Discussion:
“ Conversely, it could be inferred that four downregulated miRNAs and two upregulated miRNAs are involved in catch-up growth in SGA children (Figure 6). These miRNAs are involved in cell proliferation, cell growth, and IGF1 receptor regulation and may potentially serve as markers for the growth potential in SGA children.”
“Future studies to elucidate the downstream pathways of these miRNAs may lead to the discovery of novel target genes that are essential for child growth.”
Q 6. In future this study may be repeated in larger sample size to validate.
à Because we analyzed the miRNA profiles of only a few samples, further validation studies should be conducted to apply these results to all SGA children.
=> We have modified the last paragraph as follows:
“This study was a cross-sectional study with relatively few subjects, which makes it difficult to generalize the results to all SGA patients. Further validation studies with more subjects are needed. Longitudinal, prospective studies from birth to adulthood are needed to apply our findings to predicting catch-up growth in babies born SGA. Although the mechanism underlying catch-up growth remains unclear, we identified miRNAs involved in catch-up growth, which may help clarify the processes involved. Future studies to elucidate the downstream pathways of these miRNAs may lead to the discovery of novel target genes that are essential for child growth.”
Reviewer 2 Report
The significance and practical impact /application of the study is low. The importance in biological processes is not established /low.
How can the results be applied? If that should be of use the blood should be tested in neonatal period and predict future outcomes.
The study should be done prospectively.
Author Response
Q: The significance and practical impact /application of the study is low. The importance in biological processes is not established /low.
à Thank you for your comment. Because we analyzed the miRNA profiles in a few samples only, further validation studies should be conducted to apply these results to all SGA children.
Q: How can the results be applied? If that should be of use the blood should be tested in neonatal period and predict future outcomes. The study should be done prospectively.
à The mechanism of catch-up growth has not yet been elucidated. Although our study was conducted after the catch-up growth phase, it is still difficult to draw conclusions about whether it could be applied to predict catch-up growth. However, we have identified miRNAs involved in catch-up growth, and our result may help clarify the processes involved in catch- up growth.
Q: The study should be done prospectively.
à We agree and that is a limitation of our study. Ours was a cross-sectional study and we analyzed the miRNA profiles in a small number of samples. Longitudinal, prospective studies are necessary to validate our result. While there are practical difficulties in enrolling patients in prospective studies, we hope to study this issue while continuing to manage our patients.
=> We have modified the last paragraph as follows:
“This study was a cross-sectional study with relatively few subjects, which makes it difficult to generalize the results to all SGA patients. Further validation studies with more subjects are needed. Longitudinal, prospective studies from birth to adulthood are needed to apply our findings to predicting catch-up growth in babies born SGA. Although the mechanism underlying catch-up growth remains unclear, we identified miRNAs involved in catch-up growth, which may help clarify the processes involved. Future studies to elucidate the downstream pathways of these miRNAs may lead to the discovery of novel target genes that are essential for child growth.~
Reviewer 3 Report
The study is carried out in a systematic manner. All the methods have been described suitably. The results obtained and the author's claim agrees with each other. The results are presented in an organized way and are clear to the readers. Lastly, the Statistical analysis is appropriately performed wherever necessary.
Overall the study has high significance and the findings are important. There are no major comments or concerns at this point.
Author Response
The study is carried out in a systematic manner. All the methods have been described suitably. The results obtained and the author's claim agrees with each other. The results are presented in an organized way and are clear to the readers. Lastly, the Statistical analysis is appropriately performed wherever necessary.
Overall the study has high significance and the findings are important. There are no major comments or concerns at this point.
=> Thank you very much.
Round 2
Reviewer 1 Report
Authors fulfilled my suggestions
Reviewer 2 Report
The study must be validated in other cohorts. Not much application for the results currently.